# On the Biochemical and Physiological Responses of 'Crimson Seedless' Grapes Coated with an Edible Composite of Pectin, Polyphenylene Alcohol, and Salicylic Acid

A. A. Lo'ay [1,*], M. M. Rabie [2], Haifa A. S. Alhaithloul [3], Suliman M. S. Alghanem [4], Aly M. Ibrahim [5], Mohamed A. Abdein [6,*] and Zinab A. Abdelgawad [7]

[1] Pomology Department, Faculty of Agriculture, Mansoura University, El-Mansoura 35516, Egypt
[2] Food Industries Department, Faculty of Agriculture, Mansoura University, El-Mansoura 35516, Egypt; abomazen.egypt@gmail.com
[3] Biology Department, College of Science, Jouf University, Sakaka 42421, Saudi Arabia; haifasakit@ju.edu.sa
[4] Biology Department, Faculty of Science, Tabuk University, Tabuk 71491, Saudi Arabia; s-alghanem@ut.edu.sa
[5] Horticulture Research Institute, ARC, Giza 12619, Egypt; Dr.alyibrahim70@gmail.com
[6] Biology Department, Faculty of Art and Science, Northern Border University, Arar 97911, Saudi Arabia
[7] Botany Department, Women's College, Ain Shams University, Cairo 11566, Egypt; Zinababdelgawad@women.asu.edu.eg
* Correspondence: Loay_Arafat@mans.edu.eg (A.A.L.); abdeingene@yahoo.com (M.A.A.)

**Abstract:** The 'Crimson seedless' grape encountered several difficulties during shelf life, including weight loss, rachis browning, and berry shattering. The effect of exogenous pectin (PE) and polyphenol alcohol (PVA) with supporting salicylic acid (SA) at different concentrations (0, 1, and 2 mM) was applied. The coating was applied to bunches for 5 min and stored at room temperature ($26 \pm 1\,^\circ$C and RH $65 \pm 3$%) for 4 days. In this study, postharvest application of PE + PVA-SA can significantly reduce the cell wall degradation enzyme activities of 'Crimson seedless' grape during shelf life. 'Crimson seedless' bunches, treated with PE + PVA-SA $_{2\,\text{mmol L}^{-1}}$, had a lowered rachis browning index (RB index), weight loss (WL%), and berry shattering percentage (BS%) and preserved berry color hue angle ($h^o$) compared to untreated bunches during shelf-life duration. Moreover, the PE + PVA-SA $_{2\,\text{mM}}$ improved berry firmness (BF) and removal force (BRF). It also improved the soluble solid content (SSC%), titratable acidity (TA%), and SSC: TA-ratio, for assessing berry maturity. The cellular metabolism enzyme activities (CMEAs) of the cell wall such as polygalacturonase (PG), cellulase (CEL), xylanase (XYL), and pectinase (PT) were minimized by applying PE + PVA-SA $_{2\,\text{mM}}$ coatings throughout storage duration. The accumulation of malondialdehyde (MDA) and cell wall damage, as well as the electrolyte leakage percentage (EL%), was reduced. PE + PVA-SA $_{2\,\text{mM}}$ maintained DPPH radical quenching activities and minimized $O_2^-$ and $H_2O_2$ production rates. Collectively, these findings suggest that PE + PVA with the presence of SA as a coating treatment preserved 'Crimson seedless' bunches during shelf life. PE + PVA-SA $_{2\,\text{mM}}$ might be at least partially ascribed to the enhancement of bunches' quality traits as well as inhibiting cell wall damage during the shelf-life period.

**Keywords:** Crimson seedless; coating; shelf-life; cellular metabolism enzymes

## 1. Introduction

Grapes (*Vitis vinifera* L) are under the Vitaceae family and are one of the most widely produced fruits worldwide due to their extensive use in winemaking and as a table fruit [1]. They are highly perishable and delectable fruits with high nutritional profiles. The berries are high in fiber and folic acid, which assist in reducing body weight, blood cholesterol, and the risk of severe hypertension [2]. 'Crimson Seedless' is a late-maturing red seedless grape cultivar with firm berries. It ripens mid-September and can be stored on vines until mid-November. Moreover, it is possible to store grapes until the early winter season

under Egyptian climates [3]. Cultivars developed for sandy and reclaimed soils have been shown to be successful [4]. Grapes are grown on 77,896 hectares, with a total yield of 1,703,395 tons [5]. The 'Crimson seedless' grape faced a number of challenges, including weight loss, wilting browning, and shattered berries [6].

Fresh fruits and vegetables have a longer post-harvest life due to its edible coating. Due to its environmentally friendly nature, it is utilized in improving food appearance and providing food safety. It can be found in both animal and vegetable forms [7]. Protein, lipid, polysaccharide, or resin may be used alone or in combination as an edible covering [8]. During processing, handling, and storage, it acts as a moisture and gas barrier. Its activity or the addition of antimicrobial substances minimizes food degradation and improves safety [9]. Other benefits of edible coating include reducing packaging waste, extending the shelf life of fresh and minimally processed products, and protecting them from hazardous environmental effects by maintaining oxygen, carbon dioxide, moisture, fragrance, and taste components transmission in a food system [10]. Edible coatings, specifically in fresh fruits and vegetables, improve shelf life, limit water and moisture loss, delay the ripening process, and prevent microbial growth, according to this study [11].

Numerous research studies have been conducted in order to improve the storage life of bunches and to retain the quality of bunches by using different materials, such as edible coatings. It can be administered in a variety of ways, including dipping, misting, or packing [12]. When treating bunches, it is also used to increase their quality [13]. However, the appropriate coating has a favorable influence on many quality traits. Normally, employing proper coating, securing, and preserving the fruit is extremely important [14]. Apples, lychees, mangoes, and peaches are less likely to deteriorate [14–17]. Thus, remarkable modifications were considered in mixing two pairs of different biopolymers as a method of coating fruits, i.e., chitosan fusing with PVA or PVP plus organic acids, i.e., ascorbic, salicylic, and oxalic acid [18–20].

Pectin is a natural and helpful, renewable common polymer to use. As a result, PE is employed in a wide range of applications, including the preservation of fruit moisture, fat, and odor as well as the reduction in fruit respiration rates [21]. It reduces dryness, softens the fruit, and improves its color [22]. In addition to reducing bacterial development, the peach fruit's antioxidant system is boosted by PE application [23]. More than that, PVA has been regarded as a unique biopolymer that may be coupled with other polymers to improve coating effectiveness, most notably PE [20]. Moreover, the supplementation of salicylic acid (SA) as an anti-senescence to the mix-up enhances antioxidant performances in protecting fruits throughout shelf life [19].

Therefore, the aim of this research was to study the effect of coating by PE + PVA with SA (at different concentrations) on grapes' quality traits of 'Crimson seedless' grapes during shelf life. Moreover, its contributions to cell wall enzyme activities were assessed.

## 2. Materials and Methods

### 2.1. Experimental Setup

'Crimson seedless' vines were planted 3 × 3 m in a sandy field of a commercial farm close to the Monufia prevence in Egypt. As the SSC% in the berry juice reached 16%, the samples were harvested. Bunches (288) were picked and delivered 3 h after harvest under cooling at 13 °C. They were divided into two main batches. The first batch (144 bunches) was used for the determination of physical quality attributes, i.e., weight loss, berry shattering, rachis browning, and berry color (hue angle). This batch was divided into 4 × 36 bunches each for treatment, for which there were three replicates (e.g., 3 × 12 bunches). The second batch was used for chemical analysis and had the same fruit distribution among the treatments, as previously described. Bunches were stored separately (each sample of treatment was separate in a carton box at one layer).

*2.2. Coating Treatment Formation*

The coating materials such as PVA and PE (the PE was extracted from pectin, from citrus peel) and SA were obtained from the Sigma-Aldrich company in the St. Louis, MO, USA. All materials were of analytic quality. Deionized water has been used to produce coating treatments. PVA (3% *w/v*) was prepared and held at 80 °C for 12 h in order to stabilize. Pectin (PE; 3% *w/v*) was gradually solved by stirring. Both stocks were mixed for the polymerizing procedure for 12 h followed by using a magnetic heater with a stir at 70 °C [24]. SA was implemented with polymer PC + PVA prepared at various doses. Each dose reached one liter of the solution, and then it was kept on a magnetic heater with stirring at 25 °C for 4 h. SA was introduced towards each treatment at 0, 1, and 2 mmol L$^{-1}$. According to some similar reports [25], the possibility of adding organic acids to the polymer mixture was considered.

*2.3. Scanning Electron Microscopy (SEM) and Zeta Potential Characterization of PE + PVA*

Scanning electron microscopy was used to identify the formation of PE + PVA and its morphology. In order to assess the strength of the coating mixture, Malvern Zeta Sizer Nano ZS90 (Malvern Instruments Ltd., Great Malvern, UK) was used. In the manufacture of PENPs, the PE solution in PVA was changed from a transparent solution to a semi-turned suspension (Figure 1). These changes affected the type of PENPs that used PVAs as components. PVA-hydroxyl groups are linked to the carboxyl groups of PE during the polymerization process [26]. In Figure 1, you can observe the cross-linked PE + PVA compounds that were used to create a PE + PVA fiber. The hydrophobic surface connection group of inter-molecular and intra-molecular hydrogen chains continues to modify the communication between PE and PVA in the combination [27]. The value of ≈17.8 mV was used as a zeta potential to determine the coating mixture's endurance (Figure 2). A novel polymer, PE + PVA, was discovered to be negatively charged due to the ionization of carboxyl groups of PE [28]. Values over +30 mV or below −30 mV are an indication of stability [29].

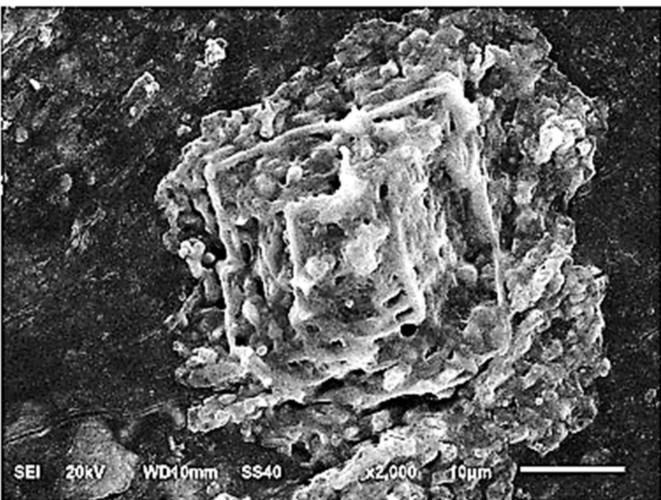

**Figure 1.** Manifests the scanning electron microscopy (SEM) image of the integrated Pectin/Polyvinyl alcohol coating that exhibited a cross-linked structure due to the formation of a Pectin/Polyvinyl alcohol network.

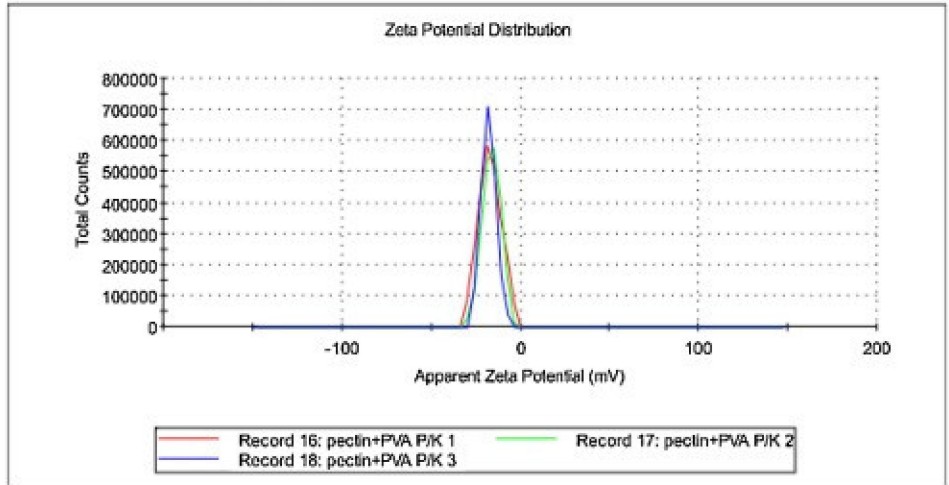

**Figure 2.** Draws the Zeta potential evaluation of Pectin/Polyvinyl alcohol polymer coating. The negative zeta potential value (−17.8 mV) indicating that the surface of the polymer is negatively charged owing to the ionization of the carboxyl groups of pectin. A zeta potential that is higher than 30 mV or lesser than −30 mV is indicative of a stable system.

### 2.4. Crimson Seedless Coating Treatment Protocol and Shelf Life

Next to the lab, 'Crimson seedless' bunches immersed in PE + PVA-SA were mixed with SA at varying doses (0, 1, and 2 mmol L$^{-1}$). We designed PE + PVA-SA treatments to achieve the following goals: untreated fruit; PE + PVA-SA $_{0\,mM}$, PE + PVA-SA $_{1\,mM}$, and PE + PVA-SA $_{2\,mM}$. The bunches were dipped in the coating solution for 5 min before being stored at room temperature (26 ± 1 °C and RH 65 ± 3%) for 4 days. Bunch samples (2 bunches) were picked every day until the end of the shelf-life duration (four days).

### 2.5. Rachis Browning (RB-Index), Berry Firmness (BF), and Separation Force (BSF)

Moreover, the rachis injury index (RII) was monitored in time. RII was characterized into five levels: RII—1, no injury; RII—2, slight symptoms; RII—3, moderate; RII—4, severe; and RII—5, very severe injury [30]. As for berry firmness and separation force, they were measured daily. Both parameters were recorded using berry texture Effegi-penetrometer supplemented with a 2 mm diameter plunger penetrator, and separation forces was estimated by utilizing a snare rather than a plunger. Firmness and separation forces of berries were presented with an N unit. Berry shattering rate and color hue angle profile were recorded [6].

### 2.6. Soluble Solids Content (SSC%), Total Acidity (TA%), SSC/TA-Ratio, and Ascorbic Acid Content (AA)

The soluble solid content (SSC%) was assessed in berry juice by using a digital refractometer (DR 6000, A. Kruss Optronic GmbH, Hamburg, Germany). TA% was estimated in 10 mL of berry juice by a titration procedure by using 0.1 N of NaOH based on tartaric acid [31]. The SSC/TA-ratio was computed from the results obtained as stated in the bunch maturity index. Ascorbic acid (AA) was assessed using the titration method by using a 2,6-dichlorophenolindophenol pigment reagent, and 6% oxalic acid was used to determine the quantity of ascorbic acid [31]. Crimson seedless berry hardness (BF) and removal force (BRF) were measured by using an Effegi penetrometer with a plunger and a 2 mm wide

penetrator, respectively. It has been reported that the BF and BRF of berries are expressed in N units. Weight loss (WL%) was assessed at the initial time of the experiment based on bunch weight and presented in percentage [30]. On the same bunches each day, the color of the berries of the 'Crimson seedless' variety (hue angle) was evaluated [6].

### 2.7. Cellular Metabolism Enzyme Activities

Berry pedicels (1 g) were ground and homogenized in a solution of 20 mM Tris-HCl at a pH of 7. This mixture was centrifuged for at $15,000 \times g$ for 20 min 4 °C. The clear supernatant was stored at −20 °C for two days in order to determine polygalacturonase (PG), xylanase (XYL), and cellulase (CEL) activities, which were monitored by using galacturonic acid, xylose, and carboxymethyl cellulose, respectively [32]. Then, 200 mL of sodium acetic acid derivation buffer (pH 5), 100 mL of sodium chloride, and 300 mL of polygalacturonic acid were added to the reaction mix (1000 mL total volume). The substrate expanded, eliciting a response. The reaction mixture was incubated in a water bath for one hour at 37 °C. Then, 500 µL dinitro salicylic acid reagent was added to the mixture, which was incubated in the water bath for 10 min. As a result, the cooled clear samples reached room temperature prior to being used. A spectrophotometer was used to determine the absorbance of the PG, XYL, and CEL mixtures at 560 and 540 nm. One unit of activity was defined as the amount that releases 1 µM of diminishing sugar per minute at 37 °C.

Pectinase activity is indicated by PT. In order to determine PT, 500 µL of 0.36% polygalacturonic acid was mixed with 0.05 M of Tris-HCL at pH 8.5, 300 µL of 4 mM $CaCl_2$, 600 µL of protein, and 600 µL of water. The findings were extremely encouraging. In order to permit reactions, the mixture was incubated at 37 °C for 3 h before measuring. In this manner, the PT could be determined by measuring the absorbance at 232 nm [33]. The activities of the enzymes are expressed in mol $s^{-1}$ $kg^{-1}$. The total protein was prepared and analyzed in order to determine the catalyst activity [34]

### 2.8. The Amount MDA Amount and EL%

Two grams of berry tissue was used to measure malondialdehyde (MDA). With the help of the TABR test, the amount of lipid peroxidation was determined. The homogenized mixture contains 2.5 g of berry tissue, 5% metaphosphoric acid (*w/v* of HPO3), and 2% butyl hydroxytoluene (BHT; $C_{15}H_{24}O$). As a result, a standard curve was prepared using 1,1,3,3-tetraethyoxypropane ($C_7H_{16}O_4$; Sigma-Aldrich, St. Louis, MO, USA), which is comparable to 0–1 mM malondialdehyde (MDA), in order to estimate MDA accumulation of kumquat peel during storage [35]. MDA was presented at a concentration of nmol $kg^{-1}$.

The samples were taken at intervals to estimate electrolyte leakage (EL) during the shelf-life period. Rachises (2 g) were added to 10 mL of 6 M mannitol and kept for 3 h in lab conditions. Next, a conductivity meter was used to measure the conductivity of the solution (M1). All cuvettes were boiled for 1 h at 100 °C to destroy the peel tissue. The conductivity of all cuvettes was then reread as total leakage (M2). Ion leakage relativity was calculated as a percentage [36].

### 2.9. $O_2^-$ and $H_2O_2$ Production Rates, as Well as DPPH* (%)

In order to determine $O_2^-$, one gram of rachis was mixed with 3 mL of $KH_2PO_4$ (50 mM, pH 7.8) at 5 °C and later filtered. The poly-vinyl-pyrrolidone (PVP 1% *w/v*) reagent was added and centrifuged at $15,000 \times g$ at 5 °C for 20 min. The observation of $NO_2$ lifespan from hydroxylamine ($H_3NO$) to creating $O_2^-$ was used to determine the rate of $O_2^-$ generation. A standard curve was used for determining $O_2^-$ production from hydroxylamine using $NO_2$. On a fresh weight basis, the $O_2^-$ value is expressed in mol $kg^{-1}$ FW [37].

The rachis sample (1 g) was mixed with 6 mL of acetone $(CH_3)_2$ CO solution in the $H_2O_2$ test. The mixture was centrifuged for 20 min at 15,000 g. The amount of 100 µL extraction was mixed with 100 µL of 5% Ti $(SO_4)_2$ and 200 µL of $NH_4OH$. After

centrifugation at $5000\times g$ for 15 min, the titanium–peroxide (TiO$_2$) complex was employed to induce a reaction, and the deposit was dissolved with 4 mL of 2 M H$_2$SO$_4$. The quantity was measured with a spectrophotometer at 415 nm, and the H$_2$O$_2$ concentration was determined [38]. The results are presented µmol kg$^{-1}$ FW based on a fresh weight basis.

The DPPH radical assay, which is based on the dismutation of radical activity, was used to determine the antioxidant activity of the rachis sample. The spectrophotometer was utilized to determine the quenching of the DPPH* by measuring the DPPH* decrease at 517 nm [39].

### *2.10. Statistical Analysis*

The experiment was designed as a randomized complete block in two-way ANOVA with two factors: PE + PVA-SA coating treatment (four levels) and shelf-life duration in days (four times) with three replicates per treatment. However, the parameters presented in Figure 1 were analyzed as a randomized complete block in one-way ANOVA when the coating treatments were a factor (measurements on the same bunches). The remaining variables were analyzed by using a factorial design. The mean separations were run with Tukey's HSD Test ($p < 0.05$). Pearson's correlation matrix among the studied parameters and principal component analysis (PCA) was applied. Tukey's HSD Test was run using the JMP Pro 16 software, with $p < 0.05$ taken as indicating a statistically significant difference (SAS Institute, Cary, NC, USA).

## 3. Results

### *3.1. Weight loss (WL%), RB-Index, Berry Shattering (BS%), and Berry Color ($h^o$)*

Figure 3 clarifies the major influence of coating application at $p < 0.05$ when the treatments and shelf-life duration in days are examined. The WL% of 'Crimson seedless' bunches over storage time is stress, which limits berry quality traits. We observed, after applying coating treatments on bunches of 'Crimson seedless,' that WL% increased independently throughout storage duration (Figure 3A). On the third day of storage, the PE + PVA-SA $_{2\,mM}$ treatment considerably reduced WL%, which was 12.46%. Despite this, the untreated bunches had a higher WL% (32.75) than 'PE + PVA-SA $_{0\,mM}$' (26.63%) and the 'PE + PVA-SA $_{1\,mM}$' (18.09%) treatments in the same period. WL% from bunches is minimized by using PE + PVA-SA $_{2\,mM}$, which provides greater maintenance for bunches during shelf life. On the third day of storage, there was a significant reduction in RB symptoms (RB-index = 1.32) compared with the other treatments (Figure 3B). The control was recorded: RB-index = 4.66 for severe injury; PE + PVA-SA $_{0\,mM}$, RB-index = 3.74 over moderate; and PE + PVA-SA $_{1\,mM}$, RB-index = 2.83 for slight symptoms. BS% showed a similar pattern to that found over the shelf life of the 'Crimson seedless' grape experiment. The lowest reduction in BS% was recorded with PE + PVA-SA $_{2\,mM}$ application on the third day of storage compared with other treatments (Figure 3C). Furthermore, PE + PVA-SA $_{2\,mM}$ had a positive effect on preserving berry color ($h^o$) in Figure 3D.

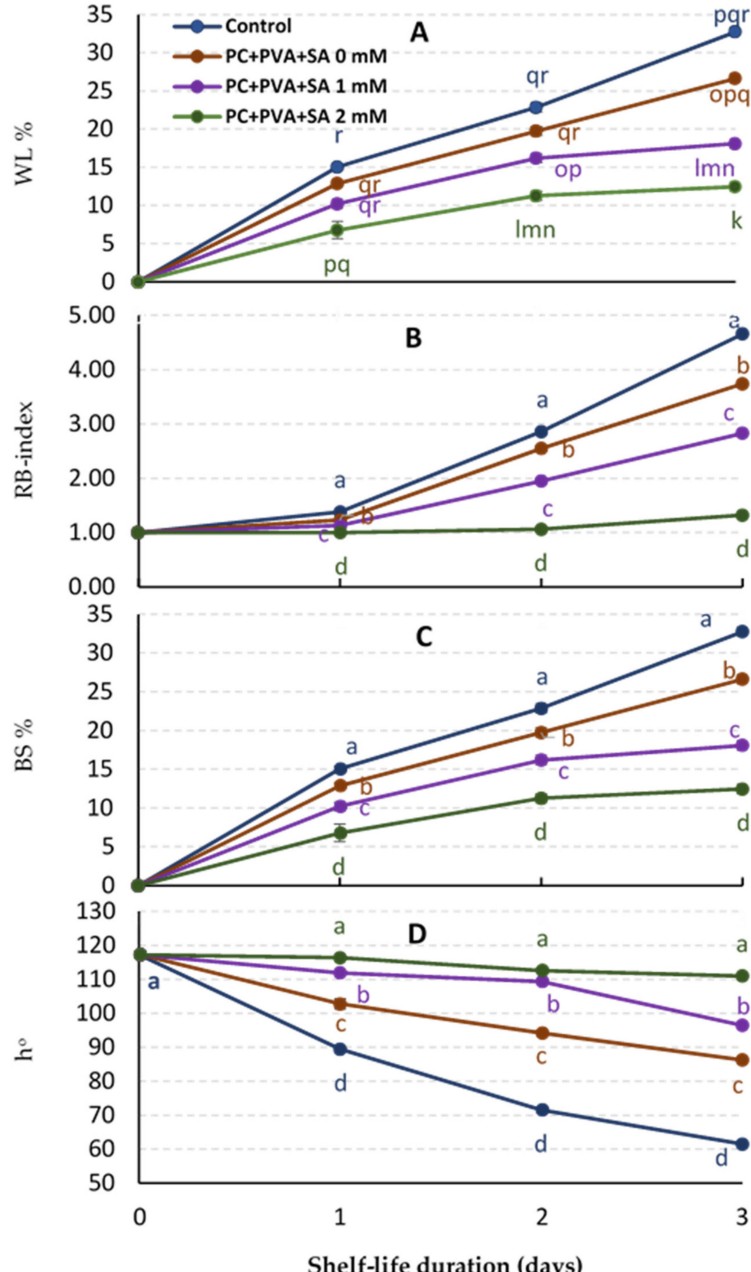

**Figure 3.** WL (%) (**A**), RB-index (**B**), BS (%) (**C**), and berry h° (**D**) of 'Crimson seedless' grapes coated with PE + PVA blending with SA at different concentrations (0, 1, and 2 mmol L$^{-1}$) and stored at room temperature (26 ± 1 °C and 65 ± 5% RH) for 3 d. Each value represents a mean of three replicates, with error bars representing standard errors (±SE), and different letters indicate significant differences at $p \leq 0.05$ between treatments. Mean comparisons were performed between all treatments.

### 3.2. BF, BRF, and AA

Figure 4 shows the results of BF, BRF, and AA. While the shelf-life lengths in days and coating applications were evaluated, the value of the parameters obtained was substantially greater at $p$ 0.05 than in earlier research. Mainly based on our data, we observed progressively diminishing BF, BRF, and AA values during the shelf-life period. The control treatment presented the lowest values than the other treatments. Consequently, the PE + PVA-SA $_{2\,mM}$ approach afforded significantly better outcomes in terms of BS (6.95 N; Figure 4A), BRF (5.75 N; Figure 4B), and AA content (1.90 mg kg$^{-1}$ FW; Figure 4C) on the

third day of shelf-life duration than in different applications and initial values. The results were remarkably similar to the control bunches (2.17 N, 2.20 N, and 0.57 mg kg$^{-1}$ FW) at the end of the experiment as opposed to the initial values (8.92, 6.19 N, and 1.97 mg kg$^{-1}$FW).

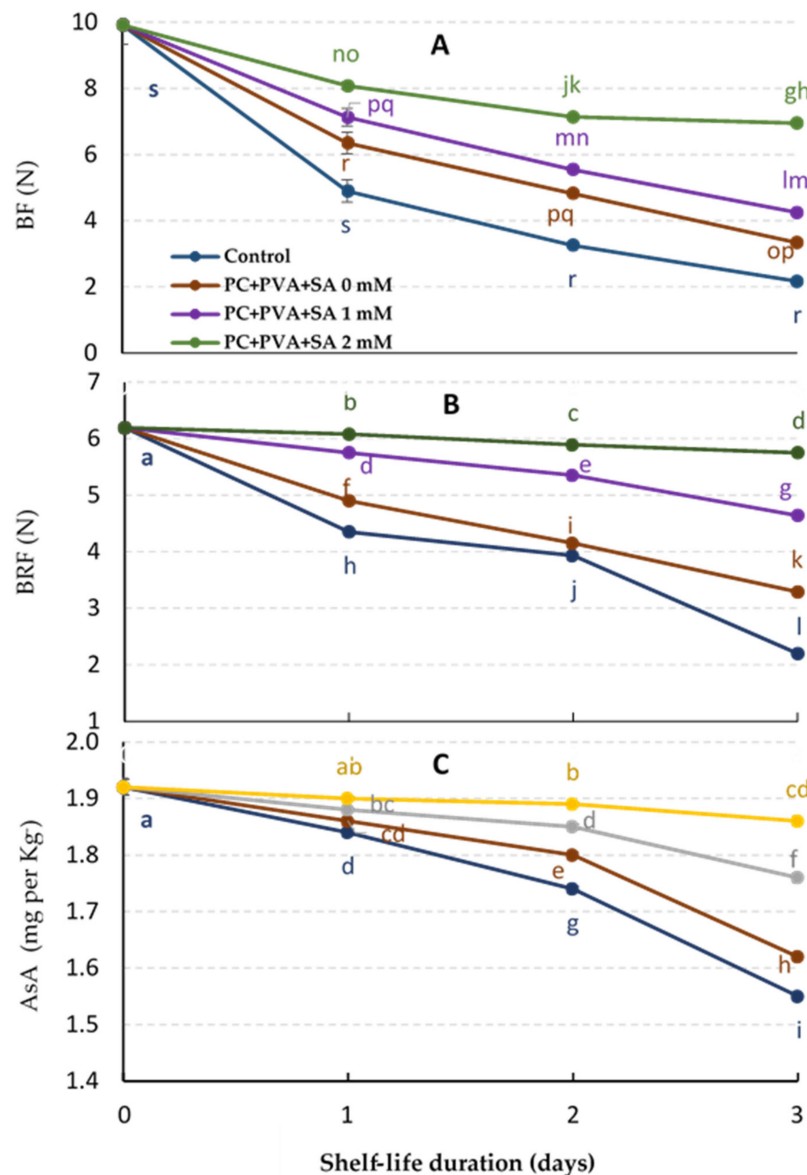

**Figure 4.** BF (**A**), BRF (**B**), and AA (**C**) of 'Crimson seedless' grapes coated with PE + PVA blending with SA at different concentrations (0, 1, and 2 mmol L$^{-1}$) and stored at room temperature (26 $\pm$ 1 °C and 65 $\pm$ 5% RH) for 3 d. Each value represents a mean of three replicates, with error bars representing standard errors ($\pm$SE), and different letters indicate significant differences at $p \leq 0.05$ between treatments. Mean comparisons were performed between all treatments.

*3.3. SSC%, TA%, and SSC: TA-Ratio*

The findings show the differences in coating treatments after 3 d of storage. Therefore, we observed a gradual increase in SSC% (Figure 5A), while TA% decreased (Figure 5B), followed by an increase in SSC: TA-ratio (Figure 5C) compared with the initial values. In addition, the PE + PVA-SA $_{2\,mM}$ treatment had a positive impact on the chemical traits of the berry. It reported the following: SSC, 16.91%; TA, 0.389%; and SSC/TA-ratio. 43.18%). On the other hand, the control bunches reported the following: SSC, 18.07%; TA, 0.214%; and SSC/TA-ratio, 84.33%.

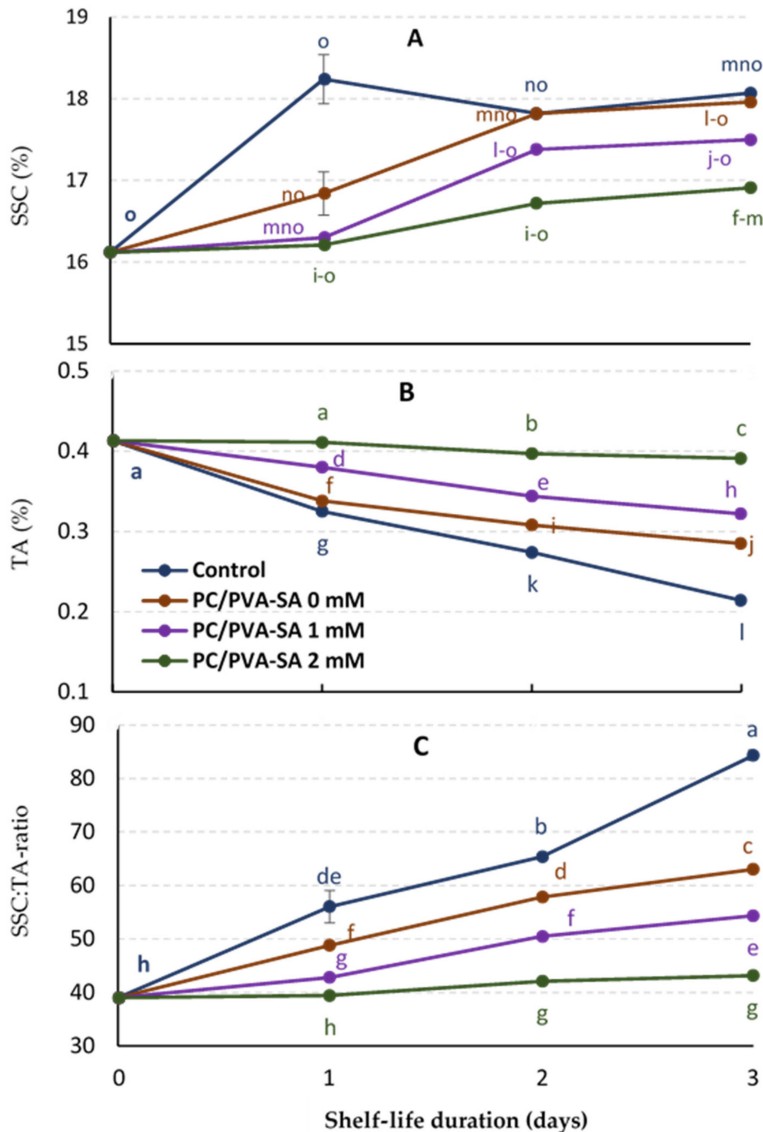

**Figure 5.** SSC (%) (**A**), TA (%) (**B**), and SSC: TA-ratio (**C**) of 'Crimson seedless' grapes coated with PE + PVA blending with SA at different concentrations (0, 1, and 2 mmol L$^{-1}$) and stored at room temperature (26 ± 1 °C and 65 ± 5% RH) for 3 d. Each value represents a mean of three replicates, with error bars representing standard errors (±SE), and different letters indicate significant differences at $p \leq 0.05$ between treatments. Mean comparisons were performed between all treatments.

### 3.4. Cellular Metabolism Enzyme Activities (CMEAs)

As shown in Figure 6, XYL, CEL, PG, and PT activations in 'Superior seedless' bunches during shelf-life changed over the course of three days. The significance at $p \leq 0.001$ indicates an interaction between storage time and coating treatments, despite the fact that both factors have been studied experimentally. As a result of 'Crimson seedless' grapes coating treatments, the enzyme activities initially slowed; they then activated differently during the storage period. The PE + PVA-SA $_{2\,mM}$ treatment has an effect on the inhibition of CMEAs during three days of shelf-life compared to other coating treatments.

All enzymes gradually increased until they reached their maximum activation peak on the third day of storage. Compared to other treatments, PE + PVA-SA $_{2\,mM}$ had the lowest activities for XYL (9.15), CEL (2.26), PG (30.24), and PT (0.61 μmol kg$^{-1}$) on the third day of shelf life. While the experiment was being conducted, it was also evident that the PE + PVA-SA $_{2\,mM}$ treatment extremely inhibited cellular enzymes.

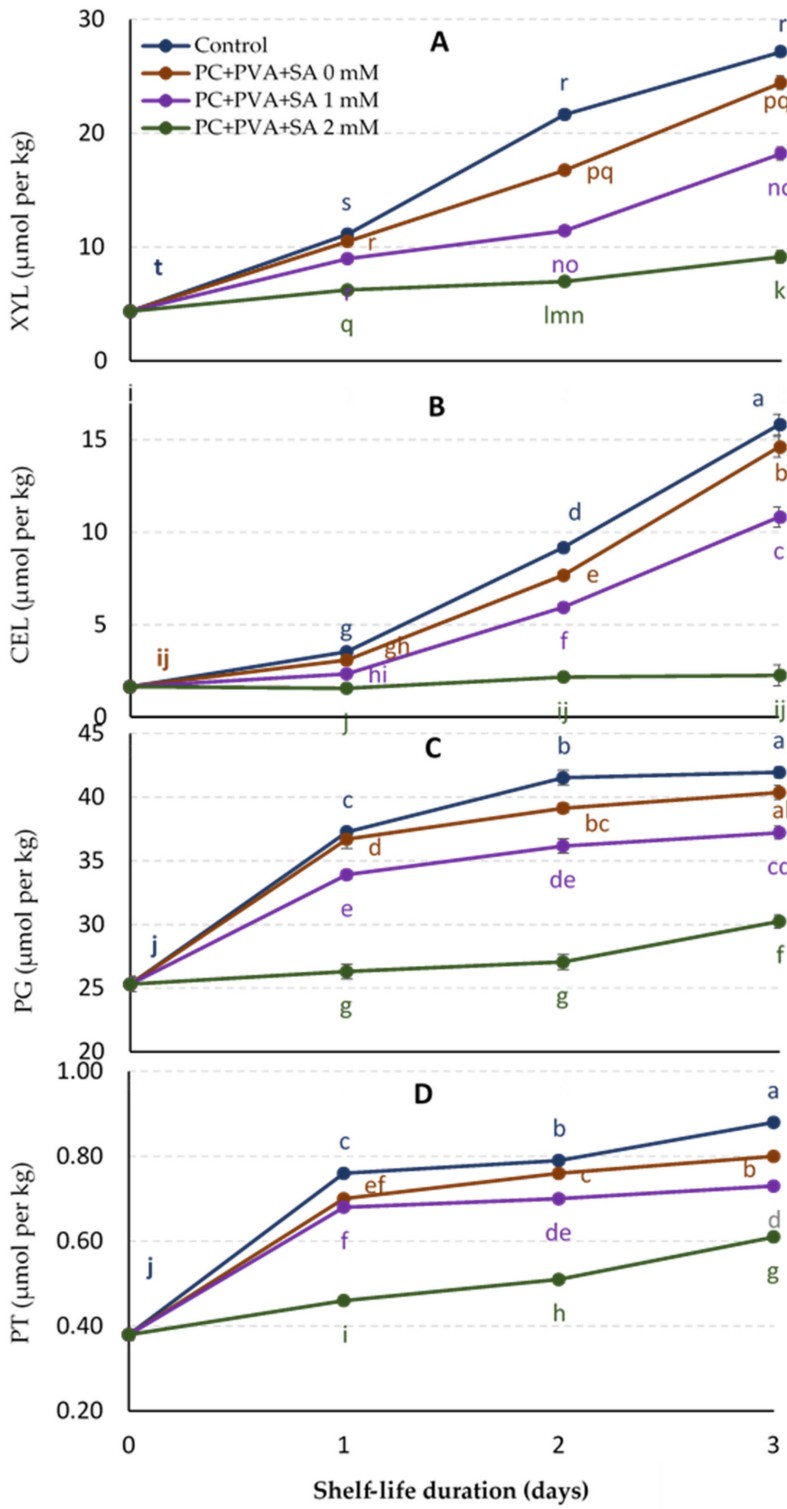

**Figure 6.** (**A**) cell wall degradation enzyme activities: xylanase (XYL), (**B**) cellulase (CEL), (**C**) polygalacturonase (PG), (**D**) and pectinase (PT) of 'Crimson seedless' grapes coated with PE + PVA blending with SA at different concentrations (0, 1, and 2 mmol L$^{-1}$) and stored at room temperature ($26 \pm 1\,°$C and $65 \pm 5\%$ RH) for 3 d. Each value represents a mean of three replicates, with error bars representing standard errors ($\pm$ SE), and different letters indicate significant differences at $p \leq 0.05$ between treatments. Mean comparisons were performed between all treatments.

### 3.5. MDA, EL%, and DPPH% Reduction

Figure 7 describes the difference in MDA accumulation and EL% and DPPH* reduction in 'Crimson seedless' grapes during shelf life. The parameters that had a positive significant interaction at $p < 0.001$ (coating application and storage duration) were studied as factors. Observably, at the initial time of the experiment, all variables increased with respect to different overall treatments. We observed that the treatment of bio-polymer PE + PVA mixed with SA at 2 mM significantly minimized MDA (Figure 7A) and EL% (Figure 7B) during storage and increased DPPH% (Figure 7C). Moreover, it lowered the accumulation of MDA, which was recorded at (0.18 mM kg$^{-1}$ FW), EL% (8.15%), and DPPH% (1.26%) on the third day of the shelf life than other treatment bunches controlled. However, the untreated treatment showed a more accelerated quantity of MDA (1.45 mM kg$^{-1}$ FW), EL% (24.45%), and DPPH% (1.11%) until the third day of storage. The presence of the highest SA concentration resulted in the lowest amounts of MDA and EL percent, while the control treatment had a low DPPH reduction percentage. The variation of MDA, EL%, and DPPH% may be due to storage time stress, which provided more accumulation in MDA, in EL%, and more activation of AEAs than compared to DPPH%.

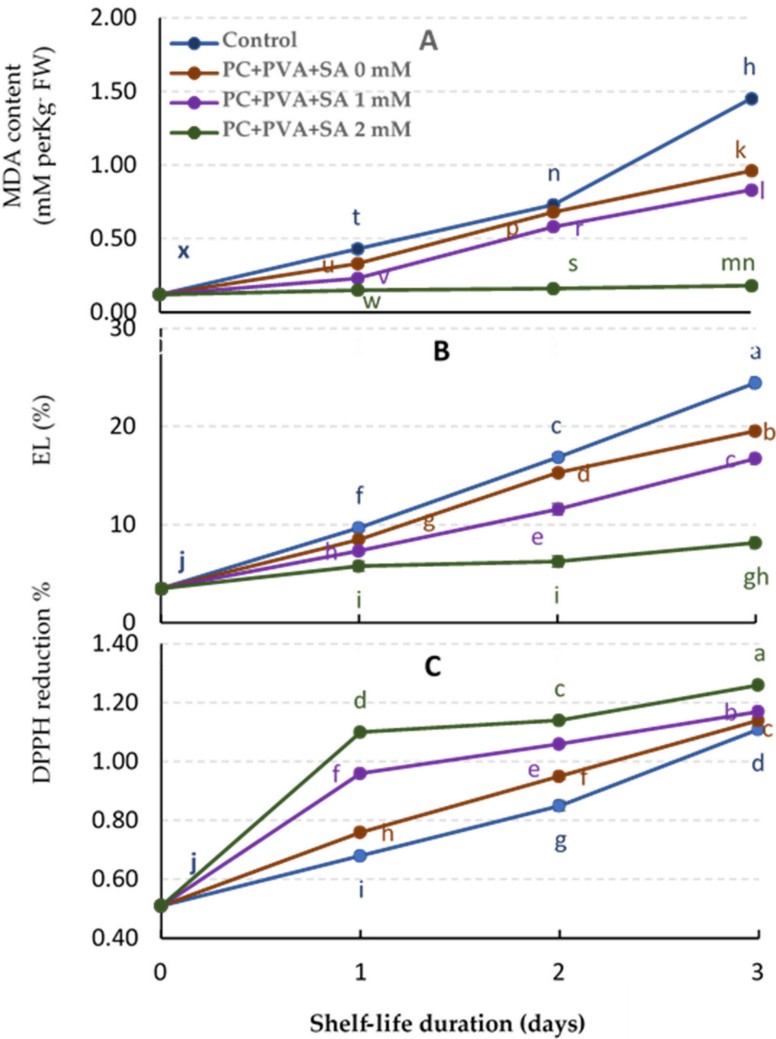

**Figure 7.** Melanodialdehyde accumulation (MDA; (**A**)), electrolyte leakage (EL%; (**B**)), and DPPH* percentage (**C**) of 'Crimson seedless' grapes coated with PE + PVA blending with SA at different concentrations (0, 1, and 2 mmol L$^{-1}$) and stored at room temperature (26 ± 1 °C and 65 ± 5% RH) for 3 d. Each value represents a mean of three replicates, with error bars representing standard errors (±SE), and different letters indicate significant differences at $p \leq 0.05$ between treatments. Mean comparisons were performed between all treatments.

### 3.6. $O_2^-$ and $H_2O_2$ Production Rate

Figure 8 presents the variations that were estimated in both $O_2^-$ and $H_2O_2$ productions throughout shelf-life duration in days. We observed that the outcomes obtained exhibit a notable interaction at $p < 0.05$ when the storage time (in days) and treatments were analyzed as experimental factors. Initially, parameter generation rates rose gradually, increasing storage time with all treatments. The highest amount of both $O_2^-$ and $H_2O_2$ (Figure 8A,B) was observed with the untreated bunches despite the other coating treatments presenting the lowest concentration according to treatments. However, the lowered values of both $O_2^-$ and $H_2O_2$ were recorded when bunches were coated with PE + PVA-SA $_{2\,mM}$ throughout for three days ($p < 0.001$; Figure 8).

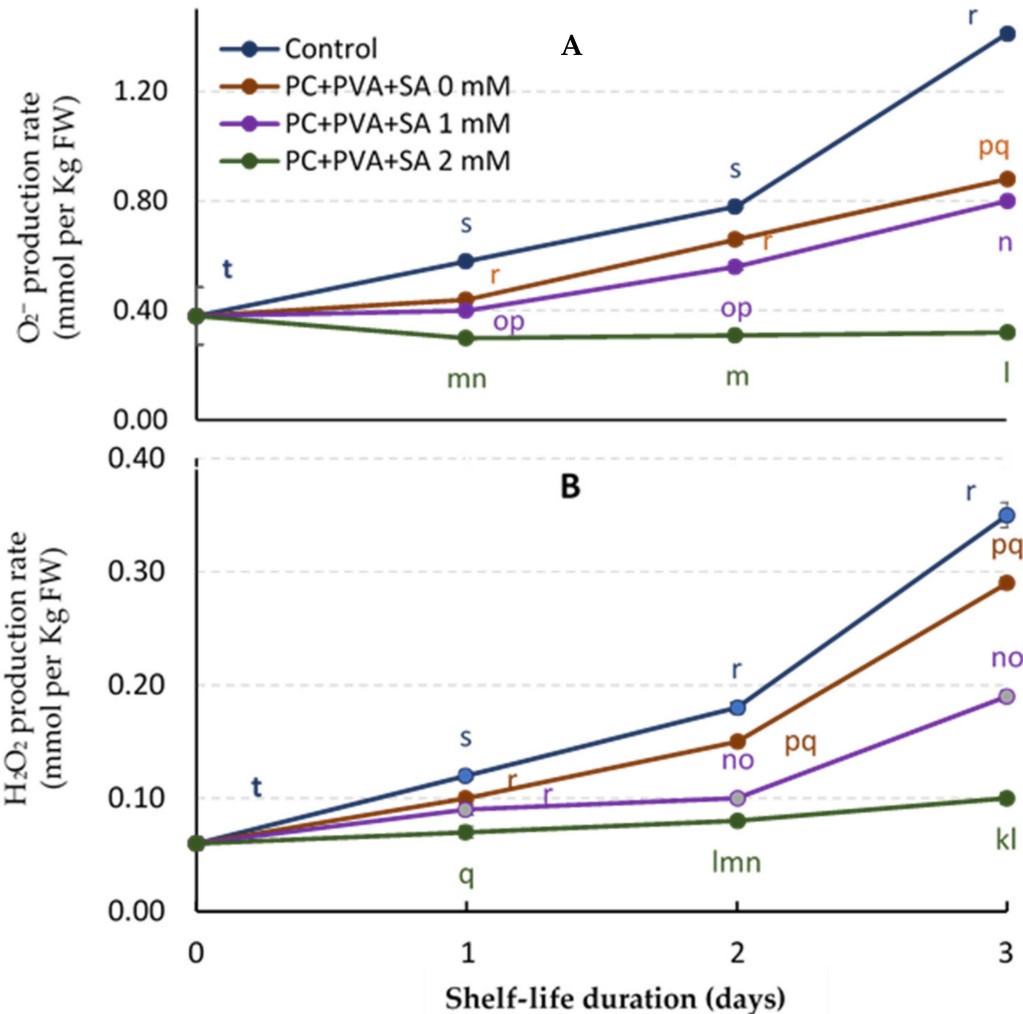

**Figure 8.** $O_2^-$ (**A**) and $H_2O_2$ (**B**) production rates of 'Crimson seedless' grapes coated with PE + PVA blending with SA at different concentrations (0, 1, and 2 mmol L$^{-1}$) and stored at room temperature (26 ± 1 °C and 65 ± 5% RH) for 3 d. Each value represents a mean of three replicates, with error bars representing standard errors (±SE), and different letters indicate significant differences at $p \leq 0.05$ between treatments. Mean comparisons were performed between all treatments.

### 3.7. Multivariate Analysis of Leaf Parameters

PCA for physiological and biochemical parameters data collected from 'Crimson seedless' bunches was carried out for different tested coating treatments (PE + PVA-SA $_{0\,mM}$, PE + PVA-SA $_{1\,mM}$, and PE + PVA-SA $_{2\,mM}$) applied and then stored for three days at room temperature. PCA separated the effect of coating treatment (PE and PVA blended with SA at different concentrations). PC1 explained 84.5% of the variability in the data, while PC2 explained 5.83% variability (Figure 9A).

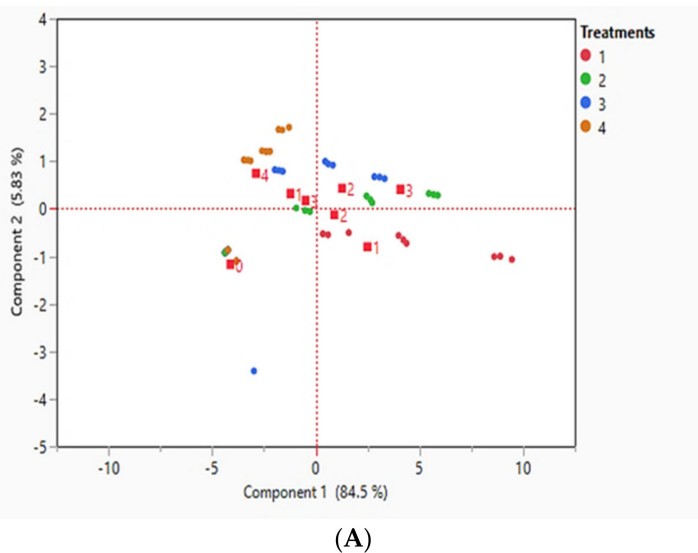
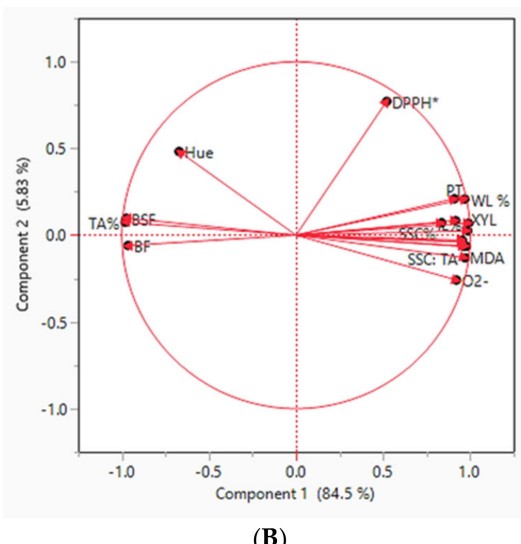

(A)  (B)

**Figure 9.** Exhibits principal component analysis (PCA) representing pectin blended with PVA supported with salicylic acid relative to 'Crimson Seedless' grape during shelf-life period, outlined with the contribution of each parameter on the two PCA axes (**A**) and all parameters measured in bunches during shelf-life duration (**B**). Principal component analysis (PCA)–variable correlation. The numbers (1–4) represent coating treatments: untreated bunches; PE + PVA-SA $_{0\,mM}$, PE + PVA-SA $_{1\,mM}$, and PE + PVA-SA $_{2\,mM}$.

Figure 9B shows the negative correlation between bunches treated by creating treatment (PE + PVA-SA) and berry quality with all variables except Pearson's correlation matrix among the studied parameters. It also shows the correlation and indicates the results (Table 1).

**Table 1.** Pearson's correlation pattern among the investigated variables of 'Crimson Seedless' grapes that are coated by four levels of mixture coating.

| | WL% | RB-Index | Hue | BS% | BF | BSF | SSC% | TA% | SSC: TA | XYL | CEL | PG | PT | MDA | IL% | DPPH* | H₂O₂ | O₂⁻ |
|---|---|---|---|---|---|---|---|---|---|---|---|---|---|---|---|---|---|---|
| WL% | *<br>1.0000 | | | | | | | | | | | | | | | | | |
| RB-Index | 0.8988 | 1.0000 | | | | | | | | | | | | | | | | |
| Hue | −0.5777 | −0.6235 | 1.0000 | | | | | | | | | | | | | | | |
| BS% | 0.8961 | 0.9838 | −0.6130 | 1.0000 | | | | | | | | | | | | | | |
| BF | −0.9525 | −0.8805 | 0.6583 | −0.8774 | 1.0000 | | | | | | | | | | | | | |
| BSF | −0.9348 | −0.9311 | 0.6676 | −0.9010 | 0.9286 | 1.0000 | | | | | | | | | | | | |
| SSC% | 0.8304 | 0.7213 | −0.5380 | 0.7135 | −0.8682 | −0.8217 | 1.0000 | | | | | | | | | | | |
| TA% | −0.9459 | −0.9227 | 0.6669 | −0.9073 | 0.9524 | 0.9824 | −0.8346 | 1.0000 | | | | | | | | | | |
| SSC: TA | 0.9230 | 0.9312 | −0.6819 | 0.9184 | −0.9206 | −0.9779 | 0.8315 | −0.9832 | 1.0000 | | | | | | | | | |
| XYL | 0.9529 | 0.9675 | −0.6346 | 0.9541 | −0.9425 | −0.9565 | 0.7900 | −0.9608 | 0.9407 | 1.0000 | | | | | | | | |
| CEL | 0.8868 | 0.9888 | −0.5981 | 0.9685 | −0.8802 | −0.9153 | 0.7197 | −0.9057 | 0.8991 | 0.9632 | 1.0000 | | | | | | | |
| PG | 0.9177 | 0.8011 | −0.5854 | 0.7900 | −0.9377 | −0.8955 | 0.8400 | −0.9367 | 0.8737 | 0.8983 | 0.8037 | 1.0000 | | | | | | |
| PT | 0.9440 | 0.7799 | −0.5484 | 0.7742 | −0.9404 | −0.8829 | 0.8391 | −0.9155 | 0.8596 | 0.8817 | 0.7779 | 0.9769 | 1.0000 | | | | | |
| MDA | 0.9164 | 0.9820 | −0.6304 | 0.9699 | −0.9102 | −0.9492 | 0.7685 | −0.9537 | 0.9576 | 0.9589 | 0.9722 | 0.8486 | 0.8330 | 1.0000 | | | | |
| IL% | 0.9644 | 0.9699 | −0.6087 | 0.9658 | −0.9419 | −0.9487 | 0.8030 | −0.9588 | 0.9443 | 0.9845 | 0.9615 | 0.8938 | 0.8898 | 0.9761 | 1.0000 | | | |
| DPPH* | 0.6632 | 0.4800 | −0.1151 | 0.5142 | −0.5270 | −0.4012 | 0.4074 | −0.4103 | 0.3863 | 0.5293 | 0.4853 | 0.4525 | 0.5712 | 0.4690 | 0.5705 | 1.0000 | | |
| H₂O₂ | 0.9094 | 0.9731 | −0.6079 | 0.9564 | −0.8668 | −0.9490 | 0.7237 | −0.9163 | 0.9315 | 0.9542 | 0.9639 | 0.7943 | 0.7940 | 0.9603 | 0.9541 | 0.4938 | 1.0000 | |
| O₂⁻ | 0.8209 | 0.9311 | −0.7148 | 0.9177 | −0.8669 | −0.9055 | 0.6818 | −0.9002 | 0.9251 | 0.8884 | 0.9098 | 0.7694 | 0.7448 | 0.9494 | 0.8938 | 0.3217 | 0.9075 | 1.0000 |

* Values represent average values per treatment and shelf-life duration in days. WL%—bunch weight loss percentage; RB Index—rachis browning index; hᵒ—hue angle of bunches; BS—berry shattering percentage; BF—berry firmness (N); BSF—berry separation force (N); SSC%—soluble solids content; TA%—total acidity; SSC: TA—ratio; XYL—xylanase; CEL—cellulase; PG—polygalacturonase; PT—pectinase; MDA—malondialdehyde accumulation; EL%—electrolyte leakage; DPPH*—scavenging activity; H₂O₂—hydrogen peroxide; O₂⁻—superoxide. The different colors of PCA values indicate the negative or positive correlations among variables.

## 4. Discussion

Berries' quality traits are important factors for consumers. In addition, the PE + PVA-SA $_{2\,mM}$ treatment tended to reduce WL% of 'Crimson seedless' bunches during shelf life. This is sufficient to increase the coating treatment capacity of waxes on the berry's surface. 'Crimson seedless' berries, on the other hand, have higher levels of wax to protect them from dehydration and pathogens during the course of their shelf life [40]. Therefore, PE + PVA-SA $_{2\,mM}$ is interpreted as minimizing water evaporation [41]. Moreover, the weight loss of bunches via transpiration and respiration is the primary cause of weight loss in fresh fruits and vegetables [42], and it minimized fungal infection [43]. Lowered RB symptoms because of the application of PE + PVA-SA $_{2\,mM}$ treatment suggested that the appearance of SA in the coat solution acts as an anti-senescence of bunch tissues [30]. Furthermore, the AsA content in 'Crimson seedless' bunches (Figure 4) is at a stable level during 4 d, which may be protected bunch tissues throughout storage by quenching ROS [44]. Therefore, RB incidences and color changes in berries may enhance dehydration processes and cell damage throughout storage duration. Bunches become increasingly susceptible to microbial infections over the shelf-life period [45]. According to earlier studies, biopolymer PE and PVA together provide better protection against dehydration of bunches throughout shelf-life length [46]. Furthermore, salicylic acid with biopolymers decreased bunch dehydration by reducing respiration [47] and repressed ethylene biogenesis [48].

All coating treatments displayed independently increased SSC contents compared to control at harvest (day 0) (Figure 5A). During 4 days of shelf life, the bunches treated with PE-PVA-SA $_{2\,mM}$ effectively retarded the degradation of SSC than control in which the dramatic increase in SSC was detected after 4 days. Our results agreed with previous studies [49]. On the other hand, the presence of SA in coating mixture treated samples significantly reduced TSS/TA deviation from initial values for up to 4 days. It could be related to SA suppressing the rate of respiration and ethylene biosynthesis [48]. Moreover, SA combined with coating blend as a treatment for bunches delayed or inhibited ripening [50]. Therefore. In term of fruits firmness, it decreased gradually during the storage period. Furthermore, during postharvest storage, bunches lost their firmness due to loss and/or changes in their structure/composition [51]. The obtained results could be due to increased inhibition occurring due to cell wall degradation enzymes activities such as PG, CEL, and XYL, which are correlated to berry firmness and removal force [52]. The mixture of PE + PVA with SA at 2 mM inhibited cell wall enzyme activities [19]. SA inhibits the activity of important cell wall degrading enzymes such as cellulase, polygalacturonase, and xylanase, preventing fruit softening [48]. Moreover, this could be related to the alteration in the structure of rachis and berry tissue throughout storage [53]. Cellulose, polygalacturonan, and xylanase are among the enzymes that may contribute to softening berry tissue [48]. PE + PVA-SA $_{2\,mM}$ decreased enzyme activity at low rates. This activity is in accordance with a prior report on the subject [53]. BF and BRF were reduced slowly and considerably more with bunches coated with PE + PVA-SA $_{2\,mM}$ compared to bunches coated with other coatings and uncoated bunches throughout storage duration. The degradation of the cell wall is reduced [54,55]. This effect could be explained by also reducing the WL% of berries and the structure of rachis tissue after storage duration. At room temperature, crimson seedless bunches are prone to weight loss and soften quickly. Fruit softening results in a loss of fruit firmness, disease infiltration, and decline of fruit quality [56,57]. In general, the action of a group of CMEAs is linked to fruit softening [58,59]. Two major enzymes that breakdown the pectin component of the cell wall are pectin methyl esterase (PME) and polygalacturonase (PG) [60]. PME catalyzes pectin demethylation, making the cell wall more vulnerable to PG breakdown [61]. Cellulase degrades xyloglucan into cellulose and hemicellulose, destroying the cell wall's pectin. In cell wall glycosyl residues, -glucosidase (-Glu) plays a critical function [60,62]. During fruit softening, the growth of water soluble pectin (WSP) and the destruction of cell wall components are also linked to a loss of firmness [62]. In order to extend the shelf life of fruits, it is necessary to suppress enzymes involved in cell wall disintegration and retain fruit firmness.

The results of this study will reveal and illuminate the impact of coating application on the chemical traits of bunches during storage. The ability to maintain the fruit by using PE + PVA-SA $_{2\,mM}$ is mostly due to the presence of SA at 2 mM, which plays the role of an anti-senescence agent, protecting tissues from oxidative stress throughout shelf life. This might result in a decrease in lipid peroxidation, protein oxidation, and cell wall damage [30]. Hence, bunches dehydrate from berry and rachis tissues [36], show the degradation process by inhibiting enzyme activities [48], and ripen late [63]. These variations might be explained by the use of PE + PVA-SA $_{2\,mM}$ as a coating on 'Crimson seedless' bunches to reduce respiration and ethylene generation rates [54].

The previous results suggest that in Figure 4, the highest amount of AsA gives the 'Crimson seedless' bunches the ability to withstand a three-day shelf-life period, which is evident from the results of this study. Due to the excessive awareness of antioxidants in bunches, they may be able to work together more effectively in a network with other antioxidants [64]. A possible explanation for the stability of cellular enzyme activities is that AsA inhibits reactive oxygen species directly, rendering the other enzymes particularly effective in quenching free radical generation [44] during the entire storage period [65]. During the storage of 'Marsh' cultivar grapes, rootstocks were found to have an impact on enzyme activities [66]. Furthermore, the outcome of the PE + PVA-SA $_{2\,mM}$ coating treatment was that it afforded more preservation quality traits of bunches. This behavior is supported by a study that reported that SA decreased CEL, XYL, PG, and PT activities (Figure 6A–C) of 'Crimson seedless' bunch tissues under shelf-life duration [67]. Moreover, it could quench reactive oxygen species throughout shelf-life [68]. According to the studies, this effect could be defined as SA is considered as a commercial chemical for supplementing fruit in order to mitigate abiotic stress [69].

It could be proposed that SA at 2 mM saved cell membranes throughout storage and, therefore, confirmed lower MDA accumulation [68]. Hence, SA inhibited CMEAs during shelf-life [69], with improved scavenging reactive oxygen species formation [70,71]. Thus, SA established a higher ability to sustain cell plasma membrane integrity throughout storage [72].

The changes in $O_2^-$ and $H_2O_2$ among coating treatments throughout shelf-life duration may confirm that SA at 2 mM decreased CMEAs, i.e., (XYL, CEL, and PG; PT Figure 6), which scavenged ROS and diminished the dismutation of $O_2^-$ and $H_2O_2$ generation. Moreover, $H_2O_2$ was quenched by activating the antioxidant network between antioxidant enzymes too [73].

## 5. Conclusions

In conclusion, the combination of two edible polymers (pectin and polyvinyl-alcohol) mixed with salicylic acid at 2 mmol $L^{-1}$ afforded more maintenance of the 'Crimson seedless' quality traits in two primary aspects: CMEAs minimized activities during shelf-life and improved the quality of bunches as a means of delaying the lipid peroxidation of cell wall damage. As a result of its CMEAs, 'Crimson seedless' grapes extended storability at room temperature for an additional four days. In the main conclusion, the combination of two edible polymers (pectin and polyvinyl-alcohol) mixed with salicylic acid at 2 mmol $L^{-1}$ afforded more maintenance of crimson seedless quality traits.

**Author Contributions:** Conceptualization, Z.A.A.; Data curation, Writing—original draft, Formal analysis, A.A.L.; Funding acquisition, H.A.S.A.; Investigation, A.M.I.; Project administration, S.M.S.A.; Software, M.M.R.; Supervision, Validation, M.A.A. All authors have read and agreed to the published version of the manuscript.

**Funding:** This research received no external funding.

**Informed Consent Statement:** Not applicable.

**Data Availability Statement:** Relevant data applicable to this research are within the paper.

**Conflicts of Interest:** The authors declare no conflict of interest.

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
