# Peer review of "On the Biochemical and Physiological Responses of ‘Crimson Seedless’ Grapes Coated with an Edible Composite of Pectin, Polyphenylene Alcohol, and Salicylic Acid"

_horticulturae, doi:10.3390/horticulturae7110498_

Round 1

Reviewer 1 Report

The manuscript has been improved substantially. The art of figures and tables should be improved again. The caption is together with figure in Fig.3, the label of letters in the figures are not separated from the inner lines. Remove lines. Graphs are better to use special software such as origin, sigmaplot.

Line 25 “PE+PVA-SA 2 mmol L-1”, while in Line 27, 31 and other places are  PE+PVA-SA 2 mM, please make them in unique.   

Author Response

The manuscript has been improved substantially. The art of figures and tables should be improved again. The caption is together with figure in Fig.3, the label of letters in the figures are not separated from the inner lines. Remove lines. Graphs are better to use special software such as origin, sigma plot.

Thank you for your comment, I agree with but to use a new program takes more or much time. Really it is very hard to do. However, I will try to improve to being clear

Line 25 “PE+PVA-SA 2 mmol L-1”, while in Line 27, 31 and other places are PE+PVA-SA 2 mM, please make them in unique.  

Corrected and unique   

Reviewer 2 Report

Dear Authors,  

The results presented are important and should be published. However, my concerns are still the following:

Lines 24-25: The importance of this study should be still explained.

Lines 39-71: In the introduction some basics information about the coatings in general are still missing. For example: what is in used now for grapes on the market?

Line 40: “Vitis vinifera” should be in italic.

Lines 75-77: The importance and the novelty of this study should be emphasized. What is new? Is this coating already in use? It has been tested before?

Lines 193-202: The test for normal distribution is still missing. For instance, Person correlation and ANOVA should be used only if the variable are normally distributed. If they are not non parametrical tests should be used.

2 mM” should be replaced with “(2mM)” in the abstract, Lines 748, 752 etc.

Overall, mine suggestion is that the manuscript would be acceptable with major revision.

Good luck!

Author Response

Dear Authors,  

The results presented are important and should be published. However, my concerns are still the following: Thank you for your comments

Lines 24-25: The importance of this study should be still explained. Explained and added

Lines 39-71: In the introduction some basics information about the coatings in general are still missing. For example: what is in used now for grapes on the market? Was added a new paragraph

Line 40: “Vitis vinifera” should be in italic. Corrected

Lines 75-77: The importance and the novelty of this study should be emphasized. What is new? Is this coating already in use? It has been tested before?

Many studies were conducted on different coating materials for different fruits, as we wrote in the introduction part. The novelty is mixing two different biopolymers (Pectin and   Polyvinyl alcohol) together to maximize the performance of coating treatment technically with grapes (sensitive to handling).

Lines 193-202: The test for normal distribution is still missing. For instance, Person correlation and ANOVA should be used only if the variable are normally distributed. If they are not non parametrical tests should be used. I was afraid to understand, but all samples were selected randomly from shelf life duration

2 mM” should be replaced with “(2mM)” in the abstract, Lines 748, 752 etc. Corrected

Overall, mine suggestion is that the manuscript would be acceptable with major revision.

Good luck! Thank you

Reviewer 3 Report

The manuscript has been revised according to previous comments, however, the manuscript is not improved significantly as there is still lacking the connection between methods, results and discussion, especially for fruits quality parameters. For example, the method of colour assessment is described in method and results, however, this quality parameter is missing from discussion section. Also,  RB index, BS% are presented in the results section, but there is no method description in materials and method, these parameter are also excluded from discussion. In addition, the results of the quality parameters of ascorbic acid (AA), colour (Hue angle), SSC, TA, SSC:TA ratio and hardness are presented in the results section, however, the parameter of ascorbic acid (AA), colour (Hue angle), TA, SSC:TA ratio, colour (Hue angle) are excluded from the discussion section.

Author Response

The manuscript has been revised according to previous comments; however, the manuscript is not improved significantly as there is still lacking the connection between methods, results and discussion, especially for fruits quality parameters. For example, the method of colour assessment is described in method and results, however, this quality parameter is missing from discussion section.

Was added

Also,  RB index, BS% are presented in the results section, but there is no method description in materials and method,

Was added

these parameter are also excluded from discussion. Was added

 In addition, the results of the quality parameters of ascorbic acid (AA), colour (Hue angle), SSC, TA, SSC:TA ratio and hardness are presented in the results section, however,  was added

the parameter of ascorbic acid (AA), colour (Hue angle), TA, SSC:TA ratio, colour (Hue angle) are excluded from the discussion section.

Was added

Reviewer 4 Report

Not all of my comments that I consider relevant have been included in the revised manuscript. The text of the article also requires editorial corrections, eg L127 digitalrefraktometer or L722 Therefore. In term of..

Detail comments:

What raw material was the pectin obtained from? It had a degree of esterification of 55% and this is the only information about the preparation used. I recommend that adding all the information that was contained in the description of the PE.

The research was conducted for three days. At what hour interval the research was carried out. Was it exactly 24 hours? The measurement should be made more precise, even up to hours, as it is of great importance in the observed changes.

Why were such parameters of fruit storage chosen, both short time, high temperature and low humidity? What were the reasons for selecting these parameters? Why was the experiment only carried out for 3 days?

The title of chapter 2.5 should be extended to the described analyzes (content of ascorbic acid, BF, BRF, WL, BS, RB)

Correct in the text and in the graphs Kg per kg

How was the WL% determined ?. The results shown in Figure 3A show that the grape weight in the control decreased by 35% during storage for 3 days. These are significant changes, in fact, almost dried fruit was obtained. Or maybe the WL% was calculated differently than it is the standard and the results show something else. I recommends clarification in the methodology and compare these results to other publications in the discussion.

Figure 9 is illegible. I recommend increasing the font size.

Author Response

Not all my comments that I consider relevant have been included in the revised manuscript. The text of the article also requires editorial corrections, e.g.  L127 digitalrefraktometer or L722 Therefore. In term of..

Both were corrected

Detail comments:

What raw material was the pectin obtained from? It had a degree of esterification of 55% and this is the only information about the preparation used. I recommend that adding all the information that was contained in the description of the PE.

Were added

The research was conducted for three days. At what hour interval the research was carried out. Was it exactly 24 hours? The measurement should be made more precise, even up to hours, as it is of great importance in the observed changes.

0 Hour (after coating treatement) until day  1   = 24 hours

day1   until day 2 =  48 hours

day 2 until day 3  = 72 hours

day 3  up to the end of day 4 = 96 hour

Why were such parameters of fruit storage chosen, both short time, high temperature and low humidity? What were the reasons for selecting these parameters? Why was the experiment only carried out for 3 days?

First, the experiment with grapes is an imitation of market conditions. Therefore, we aimed to use this mixture of biopolymers for two reasons: 1 to increase the performance of coating treatment to minimize water evaporation from bunches during marketing. 2-The addition of salicylic acid to the coating blending as an anti-repine to minimize cell wall degradation enzyme activities. Therefore, less berry shattering during marketing. Finally, minimizing water evaporation from bunches and berry shattering reduces total berry loss. Furthermore, less rachis browning. Even we consume 95% of our product in fresh. So, we defense the experiment time within four days. 

The title of chapter 2.5 should be extended to the described analyzes (content of ascorbic acid, BF, BRF, WL, BS, RB)

Corrected

Correct in the text and in the graphs Kg per kg

Corrected

How was the WL% determined ?. The results shown in Figure 3A show that the grape weight in the control decreased by 35% during storage for 3 days. These are significant changes, in fact, almost dried fruit was obtained. Or maybe the WL% was calculated differently than it is the standard and the results show something else. I recommends clarification in the methodology and compare these results to other publications in the discussion.

Sorry for tis mistake in typing we measure weight loss not water status

Figure 9 is illegible. I recommend increasing the font size.

Improved

Round 2

Reviewer 1 Report

can be considered to publish.

Reviewer 2 Report

Dear Authors,

In my opinion the manuscript could be now acceptable for publication.

Good luck!

Reviewer 3 Report

The manuscript has been revised according to previous comments, so no further revision is required.

This manuscript is a resubmission of an earlier submission. The following is a list of the peer review reports and author responses from that submission.

Round 1

Reviewer 1 Report

The manuscript reported a treatment of PE+PVA –SA to table grapes. The results included series of biochemical and physical indices to indicate the efficient of treatment on the preservation of bunched table grapes after harvest. The study takes some efforts to try to search efficient on postharvest biology taking table grapes as example. However, the manuscript does not reach the standard to be published as its stands. Some major issues should be cared and are listed below.

  • The language should be thoroughly improved. Many grammar problems and typos there. For example, the first sentence in abstract can be removed, it makes no sense here. Line 24 is an incomplete sentence. Line 25 “therefore…” is just the repetition of Line 24. Line 26 introduced the treatment method, the sentence should be reorganized. Line 32 full name of CMEA as it is first in the text. Line 43 “instrument” is better to be “treatment”, the manuscript reported a treatment method but bot instrument. O2- and H2O2 should be carefully checked the right expression form. There are many similar problems in the text.
  • The art of figures and tables should be improved, many of them are ambiguous. Fig.1 the SEM photo can be clearer, change the contrast ratio, and delete label only leave size bar there. Are Fig. 2 original results from the detector? If it is possible, re-organized it. The labels and letters are not clear enough in the manuscript, some of them are even covered by the lines. Can the authors use some special software to perform graphs? but not directly in excel, i.e. sigmaplot, origin or others is better.
  • Please check the literature carefully and make unique forms for the cited literatures.

Reviewer 2 Report

 Dear Authors,  

in the article entitled »On the Biochemical and Physiological Responses of “Crimson Seedless” Grapes Coated with Edible Composite of Pectin, Polyphenylene Alcohol, and Salicylic Acid.” the authors investigate the relevance of the coating combination of Pectin, Polyphenylene Alcohol and Salicylic Acid for “Crimson seedless” grape.

The results presented are important and should be published. However, my concerns are the following:  

Lines 24-25: The importance of this study should be explained.

Lines 47-59: The authors should shorten this part about grape variety. Instead it should be written more about the different options of coatings.

Lines 75-77: The importance and the novelty of this study should be emphasized.

Lines 229-230: The Statistical analysis should be explained in more detail. The test for normal distribution testing should be added. The PCA analysis should be described.

Lines 673-745: The capacity of the studied coating should be compared with others available on the marker or under development.

Overall, mine suggestion is that the manuscript would be acceptable with major revision. Substantial changes should be carried out before acceptance.

Good luck!

Reviewer 3 Report

This manuscript reports an experiment on “On the Biochemical and Physiological Responses of 'Crimson Seedless' Grapes Coated With an Edible Composite of Pectin, Polyphenylene Alcohol, and Salicylic Acid". This is an interesting area of work. However, there is one main issue with the manuscript where the manuscript is lacking the connection between methods, results and discussion, especially for fruits quality parameters. For example, the method of colour assessment is described in method, however, this quality parameter is missing from the results and discussion section. Also, results of the quality parameters of SSC, TA, SSC:TA ratio and hardness are presented in the results section, however, these quality parameters are excluded from the discussion section. The manuscript needs some major corrections before the manuscript can be considered as an accepted manuscript.

Points for the authors to address:

  1. Section 2.5 …..the method for quality parameters are not well described for all parameters of SSC, TA, BS, colour, etc. revision is needed for this section.
  2. Line 159 … SSC (%), TA (%), and SSC: TA-ratio. However, the content of this section is all quality parameter tested for this study. Revision needed for this section.
  3.  Line 230 and 231, the statical analysis is not  clear, whether  using JMP or SAS ?. Revision is needed for this section.
  4. Line 252  ……. AsA ….. the method of this quality parameter is not provided in the material and methods.
  5. Line 673 Discussion …. Why the quality parameters of SSC, TA, SSC:TA ratio, hardness and colour were excluded from the discussion section. These results should be discussed in the discussion section.

Reviewer 4 Report

The aim of the study was to evaluate changes in quality parameters in grapes stored for 3 days and coated with pectin, polyphenylene alcohol and various doses of salicylic acid. The subject of the manuscript is not innovative as there are many publications on the use of these substances as an edible coating. However, due to the nature of the raw material, the topic is of practical importance. The research material used in the study is very small (only three days of storage and three doses of salicylic acid), which reduces the value of the article. The study also did not plan basic research on the stored fruit (fungal infection, respiration rate, ethylene synthesis), which also reduces the value of the study. The analytical methods used are also very basic and there are no chromatographic methods to determine the content of individual compounds.

Detail comments:

In the abstract, I recommend not to include all the markings that were made on the fruit, as their description takes almost 3/4 of the text. It is more important to write the most important results obtained in the abstract.

The methodological part requires supplements. What raw material was the pectin obtained from? It had a degree of esterification of 55% and this is the only information about the preparation used. I recommend that adding all the information that was contained in the description of the PE.

The research was conducted for three days. At what hour interval the research was carried out. Was it exactly 24 hours? The measurement should be made more precise, even up to hours, as it is of great importance in the observed changes.

Why were such parameters of fruit storage chosen, both short time, high temperature and low humidity? What were the reasons for selecting these parameters? Why was the experiment only carried out for 3 days?

L158p3 – none ±

The title of chapter 2.5 should be extended to the described analyzes (content of ascorbic acid, BF, BRF, WL, BS, RB)

The text of the article should be checked by a native speaker, as it contains incomprehensible phrases eg. L176p4 “supernatant was stored at -20°C for a period of xylanase”. L187p4 “Indicated by pectinase activity”.

The description of enzyme determination methods is very incomprehensible and unclear. I recommend improving the analytical methods for determining enzyme activity.

Correct in the text and in the graphs Kg per kg

How was the WL% determined ?. The results shown in Figure 3A show that the grape weight in the control decreased by 35% during storage for 3 days. These are significant changes, in fact, almost dried fruit was obtained. Or maybe the WL% was calculated differently than it is the standard and the results show something else. I recommends clarification in the methodology and compare these results to other publications in the discussion.

A very large number of abbreviations are used in the manuscript, which makes the text very difficult to understand. What do the abbreviations Na +%, Cl-% mean and where is the method of determining these parameters described. The correlation between chlorophyll a and b with total chlorophyll is also described, while these results are not shown. This requires some explanation. What do the abbreviations ChlA: B, Caro and Fv / Fm mean? And why are these parameters not in the Pearson correlation table?

What does mean “rchi” in the caption of table 1

In the discussion it is said that the ascorbic acid content is stable when stored for 3 days (L684). However, it can be seen from graph number 4 that there is a slight statistical change in content during the 3 days of storage. I recommend limiting such statements if there is a statistical decrease in content of ascorbic acid.

Figure 9 is illegible. I recommend increasing the font size.